# Learning Treatment Effects in Panels with General Intervention Patterns

**Vivek F. Farias**
Sloan School of Management
MIT
Cambridge, MA 02139
vivekf@mit.edu

**Andrew A. Li**
Tepper School of Business
Carnegie Mellon University
Pittsburgh, PA 15213
aali1@cmu.edu

**Tianyi Peng**
Department of Aeronautics and Astronautics
MIT
Cambridge, MA 02139
tianyi@mit.edu

## Abstract

The problem of causal inference with panel data is a central econometric question. The following is a fundamental version of this problem: Let $M^*$ be a low rank matrix and $E$ be a zero-mean noise matrix. For a 'treatment' matrix $Z$ with entries in $\{0, 1\}$ we observe the matrix $O$ with entries $O_{ij} := M^*_{ij} + E_{ij} + \mathcal{T}_{ij} Z_{ij}$ where $\mathcal{T}_{ij}$ are unknown, heterogenous treatment effects. The problem requires we estimate the average treatment effect $\tau^* := \sum_{ij} \mathcal{T}_{ij} Z_{ij} / \sum_{ij} Z_{ij}$. The synthetic control paradigm provides an approach to estimating $\tau^*$ when $Z$ places support on a single row. This paper extends that framework to allow rate-optimal recovery of $\tau^*$ for general $Z$, thus broadly expanding its applicability. Our guarantees are the first of their type in this general setting. Computational experiments on synthetic and real-world data show a substantial advantage over competing estimators.

## 1 Introduction

Consider that we are given a sequence of $T$ observations on each of $n$ distinct 'units'. In an econometric context, a unit might correspond to a geographic region with the associated sequence of observations corresponding to some economic time series of interest. In e-commerce, a unit may correspond to a customer, with the associated sequence of observations corresponding to site-visits for that customer over time. For each unit, some subset of its observations are potentially impacted by an intervention. This may correspond to a new economic policy in the econometric example, or the application of a new type of promotion in the e-commerce context. The question at hand is to estimate the 'average treatment effect' of this intervention. This is a core question in econometrics.

The problem above can be formalized as follows: Let $M^* \in \mathbb{R}^{n \times T}$ be a fixed, unknown matrix and $E$ be a zero-mean random matrix; we refer to $M^* + E$ as the 'counterfactual' matrix with each row corresponding to a distinct 'unit'. A known 'treatment' matrix $Z$ with entries in $\{0, 1\}$ encodes observations impacted by an intervention. Specifically, we observe a matrix $O$ with entries $O_{ij} = M^*_{ij} + E_{ij} + \mathcal{T}_{ij} Z_{ij}$ where the $\mathcal{T}_{ij}$ are unknown, heterogenous treatment effects.[1] Our goal is to recover the average treatment effect $\tau^* = \sum_{ij} \mathcal{T}_{ij} Z_{ij} / \sum_{ij} Z_{ij}$.

---

[1] Observable covariates on each unit may also be available; we suppress this aspect here for clarity.

35th Conference on Neural Information Processing Systems (NeurIPS 2021).

The 'synthetic control' paradigm [1, 2] has been a transformative approach to addressing this problem. That setup addresses the special case of this problem where a single unit (say, the first row) is treated; i.e. $Z$ has support on a portion of the first row. The approach is conceptually simple: find a linear combination of the untreated units (i.e. all rows other than the first) that approximate the first row on its *untreated* entries. One then uses the same linear combination to impute counterfactual values i.e. the entires of $M^*$ in the support of $Z$. Under the assumption that $M^*$ is low rank (and other assumptions), one can show that this approach recovers $\tau^*$.

Applications like the e-commerce setup alluded to earlier require we allow for more general intervention patterns, i.e. allow for *general $Z$* as opposed to the single row (or block) support required in synthetic control. A tantalizing possibility, first raised by [3], is treating entries of $M^*$ in the support of $Z$ as missing and applying matrix completion techniques to impute these counterfactual values. This viewpoint has the benefit of making no assumptions on the treatment effects and yields algorithms that do not require any special structure for $Z$. However, it is unclear that counterfactual recovery – which is equivalent to matrix recovery with general missingness patterns – is actually possible via such a method, so that recovery guarantees are typically unavailable.

## 1.1 This Paper

Succinctly, we develop an estimator that recovers the average treatment effect under provably minimal assumptions on $Z$. This is made possible by a new de-biasing identity and an extension of the entry-wise uncertainty quantification analysis of [4, 5] to general non-random missingness patterns that is of independent interest.

In providing context for our contributions, it is worth asking what one can hope for in this problem. In addition to requiring that $M^*$ have low rank (say $r$), it is clear that we *cannot* in general expect to recover $\tau^*$ absent assumptions on the treatment matrix $Z$. For instance, we must rule out the existence of a rank $r$ matrix $M'$, distinct from $M^*$ for which $M' = M^* + \gamma Z$ for some $\gamma \neq 0$, or else identifying $\tau^*$ is impossible even if $E$ is identically zero (we could rule this out if $Z$ were not in the tangent space of $M^*$). Separately, unlike matrix completion, we actually *observe* $M_{ij}^* + \mathcal{T}_{ij} + E_{ij}$ on treated entires. If the heterogeneity in treatment effects is too large, however, it is unclear that these observations are of much value. Thus, in addition to benign assumptions on $M^*$ and $E$, we make a set of assumptions on (A) the projection of $Z$ onto the tangent space of $M^*$ and (B) the heterogeneity in treatment effects. Against this backdrop, we make the following contributions:

*Rate Optimal Estimator:* We construct an estimator that achieves rate optimal guarantees for the recovery of the average treatment effect $\tau^*$ (Theorem 1) with general treatment patterns. We show under additional assumptions that our estimator is asymptotically normal (Theorem 2).

*Minimality of Assumptions:* Should the conditions we place on the projection of $Z$ onto the tangent space of $M^*$ be violated by an amount that grows small with problem size, we show *no algorithm* can recover $\tau^*$ even with homogeneous treatments (Proposition 2). Our assumptions on heterogeneity are also shown to be minimal, and satisfied by extant models in the synthetic control literature.

*General Treatment Patterns Work:* We show that our assumptions on $Z$ are satisfied by general treatment patterns, e.g., requiring that at least a constant fraction of entries are *not* treated or matrices $Z$ with rank polynomially larger than the rank of $M^*$.

*Empirical Performance:* We show both for synthetic and real data that our estimator provides a material improvement in empirical performance relative to available alternatives, including matrix completion based estimators and, where applicable, state-of-the-art synthetic control estimators.

## 1.2 Related Literature

The synthetic control literature pioneered by [1, 2] has grown to encompass sophisticated learning and inferential methods; see [6] for a review. [7, 8, 9] consider a variety of regularized regression techniques to learn the linear combination of untreated units that yields a synthetic control. [10, 11, 12] consider instead the use of principal component regression techniques. [13] propose alternative approaches to imputing counterfactuals by averaging across both untreated units (rows) and time (untreated columns). [14, 15] address inferential questions that arise in synthetic control with the latter providing a permutation test that is generally applicable.

Matrix completion methods present a means to allow for inference with *general* treatment patterns. [3] are among the first to study this, but provide no guarantees on recovering the average treatment effect. Alternatively methods that do provide guarantees on the recovery of treatment effects via matrix completion tend to make strong assumptions: [16] effectively assume that $Z$ has support on a block (so that traditional synthetic control techniques could also apply), [17] make stationarity assumptions on $M^*$ and require it to be zero-mean and [18] assume that $Z$ has i.i.d. entries (wherein a trivial estimator of the average treatment effect is also applicable).

A distinct (and common) setting for treatment effect estimation presents us with a single observation per unit and a rich set of observed covariates on these units. These observed covariates roughly ensure exchangeability of units across interventions within covariate strata. ML has been broadly used in this setting; eg. [19, 20, 21]. In contrast for our problem, one could view these covariates as latent.

Our estimation procedure begins with first computing a 'rough' estimate of the treatment effect via a natural convex estimator; [22, 23, 24] are empirical studies that have uses this estimator. Crucially, we provide a new de-biasing technique that allows for recovery guarantees and exhibits a significant performance improvement relative to this rough estimate. It is also worth noting that this convex estimator also finds application in the related problem of panel data regression; see [25, 26, 27]. State of the art methods there effectively require that $Z$ is dense.

Whereas work on matrix completion with non-standard observation patterns [28, 29, 30, 31] exists this is by and large not obviously useful or applicable to our problem. Instead, we build on a recent program to bridge convex and non-convex formulations for matrix completion [4] and Robust-PCA [5]. That work has provided a pioneering analysis of entry-wise guarantees and uncertainty quantification for convex matrix completion estimators wherein entries remain missing at random. Our work here may be viewed as extending that program to a broad class of non-random missingness patterns, a contribution of important independent interest.

## 2   Model and Algorithm

We begin by formally defining our problem; we in fact present a generalization to the problem described in the previous section, allowing for *multiple* treatments. Let $M^* \in \mathbb{R}^{n \times n}$ be a fixed rank-$r$ matrix with singular value decomposition (SVD) denoted by $M^* = U^* \Sigma^* V^{*\top}$, where $U^*, V^* \in \mathbb{R}^{n \times r}$ have orthonormal columns, and $\Sigma^* \in \mathbb{R}^{r \times r}$ is diagonal with diagonal entries $\sigma_1 \geq \cdots \geq \sigma_r > 0$.[2] Let $\sigma_{\max} := \sigma_1, \sigma_{\min} := \sigma_r$ and $\kappa := \sigma_{\max}/\sigma_{\min}$ be the condition number of $M^*$. There are $k$ treatments that can be applied to each entry, and for each treatment $m \in \{1, \dots, k\}$, we are given a *treatment matrix* $Z_m \in \{0, 1\}^{n \times n}$ which encodes the entries which have received the $m^{\text{th}}$ treatment (0 meaning no treatment, and 1 meaning being treated).[3] We then observe a single matrix of *outcomes*:

$$O := M^* + E + \sum_{m=1}^k \mathcal{T}_m \circ Z_m$$

($\circ$ is the Hadamard or 'entrywise' product), where each $\mathcal{T}_m \in \mathbb{R}^{n \times n}$ is an unknown matrix of *treatment effects*, and $E \in \mathbb{R}^{n \times n}$ is a (possibly heterogeneous) random noise matrix. Finally, let $\tau^* \in \mathbb{R}^k$ be the vector of *average treatment effects*, whose $m^{\text{th}}$ value is defined as $\tau_m^* := \langle \mathcal{T}_m, Z_m \rangle / \|Z_m\|_1$ and let $\delta_m = \mathcal{T}_m \circ Z_m - \tau_m^* Z_m$ be associated 'residual' matrices. Our problem is to estimate $\tau^*$ after having observed $O$ and $Z_1, \dots, Z_k$.

It is worth noting that the representation above is powerful: for instance, it subsumes the setting where the intervention on any entry is associated with a $\{0, 1\}^k$-valued covariate vector, and the treatment effect observed on that entry is some linear function of this covariate vector plus idiosyncratic noise. Recovery of $\tau^*$ is then equivalent to recovering covariate dependent heterogeneous treatment effects.

Our problem also subsumes the synthetic control setting where $k = 1$ and $Z_1$ must place support on a single row; the focus of our later analysis will be the case where $Z_1$ is allowed to be general.

---

[2]Note that in contrast to the previous section, we are now assuming *square* matrices (i.e. $n = T$ in the notation of the previous section). This is purely to simplify the notation – for a rectangular $n$-by-$T$ matrix, all of our theoretical guarantees hold if one swaps $n$ with $\min\{n, T\}$.

[3]We allow for multiple treatments to be applied to an entry.

The assumptions that we will need to impose in order to state meaningful results can be divided into two groups. The first are assumptions on $M^*$ and $E$ that are, by this point, canonical in the matrix completion literature:

*Assumption* 1 (Random Noise). The entries of $E$ are independent, mean-zero, sub-Gaussian random variables with sub-Gaussian norm bounded by $\sigma$: that is, $\|E_{ij}\|_{\psi_2} \leq \sigma$ for every $i, j \in [n]$.

*Assumption* 2 (Incoherence). $M^*$ is $\mu$-incoherent:

$$\|U^*\|_{2,\infty} \leq \sqrt{\mu r/n} \quad \text{and} \quad \|V^*\|_{2,\infty} \leq \sqrt{\mu r/n},$$

where $\|\cdot\|_{2,\infty}$ denotes the maximum $\ell_2$-norm of the rows of a matrix.

In addition to these standard conditions on $M^*$ and $E$ which we will assume throughout this paper, we will also need to impose conditions on the relationship between $M^*$ and the $Z_m$'s. Loosely speaking, these conditions preclude treatment matrices $Z_m$ that can be 'disguised' within $M^*$, in the sense that their projections onto the tangent spaces of $M^*$ are large. Specifically, the formal statements relate to a particular decomposition of the linear space of $n \times n$ matrices, $\mathbb{R}^{n \times n} = T^* \oplus T^{*\perp}$, where $T^*$ is the tangent space of $M^*$ in the manifold consisting of matrices with rank no larger than $\mathrm{rank}(M^*)$:

$$T^* = \{U^* A^\top + B V^{*\top} \mid A, B \in \mathbb{R}^{n \times \mathrm{rank}(M^*)}\}.$$

Equivalently, the orthogonal space of $T^*$, denoted $T^{*\perp}$, is the subspace of $\mathbb{R}^{n \times n}$ whose columns and rows are orthogonal, respectively, to the spaces $U^*$ and $V^*$. Let $P_{T^{*\perp}}(\cdot)$ denote the projection operator onto $T^{*\perp}$:

$$P_{T^{*\perp}}(A) = (I - U^* U^{*\top}) A (I - V^* V^{*\top}).$$

We will defer the formal statements of the additional conditions to the next section, but suffice to say for now that they assume $P_{T^{*\perp}}(Z_m)$ to be sufficiently large (we will show that lower bounds on the size of $P_{T^{*\perp}}(Z_m)$, in the the precise form of our own conditions, are nearly necessary). In the remainder of this section, we will outline the core contribution of this paper, which is an estimator for $\tau^*$ with a provably rate-optimal guarantee.

## 2.1 A De-biased Convex Estimator

Our estimator is constructed in two steps, stated as Eqs. (1a) and (1b) below:

$$(\hat{M}, \hat{\tau}) \in \operatorname*{argmin}_{M \in \mathbb{R}^{n \times n}, \tau \in \mathbb{R}^k} \quad g(M, \tau) := \frac{1}{2} \left\| O - M - \sum_{m=1}^{k} \tau_m Z_m \right\|_{\mathrm{F}}^2 + \lambda \|M\|_*, \qquad (1\mathrm{a})$$

$$\tau^d := \hat{\tau} - D^{-1}\Delta. \qquad (1\mathrm{b})$$

In Eq. (1b), define by $D \in \mathbb{R}^{k \times k}$ the Gram matrix with entires $D_{lm} = \langle P_{\hat{T}^\perp}(Z_l), P_{\hat{T}^\perp}(Z_m)\rangle$, and by $\Delta \in \mathbb{R}^k$ the 'error' vector with components $\Delta_l = \lambda \langle Z_l, \hat{U}\hat{V}^\top \rangle$, where we have let $\hat{M} = \hat{U}\hat{\Sigma}\hat{V}^\top$ be the SVD of $\hat{M}$, and let $\hat{T}$ denote the tangent space of $\hat{M}$.[4]

The first step, Eq. (1a), is a natural convex optimization formulation that we use to compute a 'rough' estimate of the average treatment effects. The objective function's first term penalizes choices of $M$ and $\tau$ which differ from the observed $O$, and the second term seeks to penalize the rank of $M$ using the nuclear norm as a (convex) proxy. The tuning parameter $\lambda > 0$, which will be specified in our theoretical guarantees, encodes the relative weight of these two objectives.

After the first step, having $(\hat{M}, \hat{\tau})$ as a minimizer of Eq. (1a), we could simply use $\hat{\tau}$ as our estimator for $\tau^*$. However, a brief analysis of the first-order optimality conditions for (1a) yields a simple, but powerful decomposition of $\hat{\tau} - \tau^*$ that suggests a first-order improvement to $\hat{\tau}$ via *de-biasing*:

**Lemma 1.** *Suppose* $(\hat{M}, \hat{\tau})$ *is a minimizer of* (1a). *Let* $\hat{M} = \hat{U}\hat{\Sigma}\hat{V}^\top$ *be the SVD of* $\hat{M}$, *and let* $\hat{T}$ *denote the tangent space of* $\hat{M}$. *Denote* $\hat{E} = E + \sum_m \delta_m \circ Z_m$. *Then,*

$$D(\hat{\tau} - \tau^*) = \Delta^1 + \Delta^2 + \Delta^3, \qquad (2)$$

*where* $\Delta^1, \Delta^2, \Delta^3 \in \mathbb{R}^k$ *are vectors with components*

$$\Delta_m^1 = \lambda\langle Z_m, \hat{U}\hat{V}^\top\rangle, \Delta_m^2 = \langle P_{\hat{T}^\perp}(Z_m), \hat{E}\rangle, \Delta_m^3 = \langle Z_m, P_{\hat{T}^\perp}(M^*)\rangle.$$

---

[4]Our definition implicitly assumes that $D$ is invertible. We view this as a natural assumption on (the absence of) collinearity in treatments.

Consider this error decomposition, i.e. $\hat{\tau} - \tau^* = D^{-1}(\Delta^1 + \Delta^2 + \Delta^3)$ by Eq. (2), and note that $D^{-1}\Delta^1$ is entirely a function of observed quantities. Thus, it is known and *removable*. The second step of of our algorithm, Eq. (1b), does exactly this. The resulting de-biased estimator, denoted $\tau^d$, is the subject of this paper. As an aside, it is worth noting that while de-biased estimators for high-dimensional inference have received considerable attention recently, our de-biasing procedure is algorithmically distinct from existing notions of de-biasing, including those for problems closely related to our own (e.g. matrix completion [32, 18] and panel data regression [27]).

Our main results characterize the error $\tau^d - \tau^*$. The crux of this can be gleaned from the second and third terms of Eq. (2). If $\hat{T}$ is sufficiently 'close' to $T^*$, then $\Delta^3$ becomes negligible (because $P_{T^{*\perp}}(M^*) = 0$). Showing closeness of $\hat{T}$ and $T^*$ is the main technical challenge of this work. The remaining error, contributed by $\Delta^2$, can then be characterized as a particular 'weighted average' of the (independent) entries of $E$ and the residual matrices $\delta_m$ which we show to be min-max optimal.

To conclude this section, we prove Lemma 1 for a single treatment ($k = 1$); the complete proof is a straightforward generalization, completed in Appendix A (See the online full version[5] for Appendix details).

*Proof of Lemma 1 for $k = 1$.* Since $k = 1$, we suppress redundant subscripts. Consider the first-order optimality conditions of (1a):

$$\left\langle Z, O - \hat{M} - \hat{\tau}Z \right\rangle = 0 \tag{3a}$$

$$O - \hat{M} - \hat{\tau}Z = \lambda(\hat{U}\hat{V}^\top + W), \tag{3b}$$

$$\|W\| \leq 1 \tag{3c}$$

$$P_{\hat{T}^\perp}(W) = W \tag{3d}$$

($W$ is called the 'dual certificate' in the matrix completion literature [33, 34, 35]). Combining Eq. (3b) and Eq. (3a), we have

$$\left\langle Z, O - \hat{M} - \hat{\tau}Z \right\rangle = 0 \implies \left\langle Z, \lambda(\hat{U}\hat{V}^\top + W) \right\rangle = 0 \implies \lambda\left\langle Z, \hat{U}\hat{V}^\top \right\rangle = -\left\langle Z, \lambda W \right\rangle. \tag{4}$$

Next, applying $P_{\hat{T}^\perp}(\cdot)$ to both sides of Eq. (3b) and using Eq. (3d):

$$P_{\hat{T}^\perp}(O - \hat{M} - \hat{\tau}Z) = \lambda W$$
$$\implies P_{\hat{T}^\perp}(M^*) + P_{\hat{T}^\perp}(E + \delta \circ Z) - (\hat{\tau} - \tau^*)P_{\hat{T}^\perp}(Z) = \lambda W, \tag{5}$$

where the implication is by definition: $O = M^* + E + \tau^*Z + \delta \circ Z$. Finally, substituting $\lambda W$ from Eq. (5) into Eq. (4), we obtain

$$\lambda\left\langle Z, \hat{U}\hat{V}^\top \right\rangle = -\left\langle Z, P_{\hat{T}^\perp}(M^*) + P_{\hat{T}^\perp}(E + \delta \circ Z) - (\hat{\tau} - \tau^*)P_{\hat{T}^\perp}(Z) \right\rangle$$

$$\implies (\hat{\tau} - \tau^*)\left\| P_{\hat{T}^\perp}(Z) \right\|_{\mathrm{F}}^2 = \lambda\left\langle Z, \hat{U}\hat{V}^\top \right\rangle + \left\langle Z, P_{\hat{T}^\perp}(E + \delta \circ Z) \right\rangle + \left\langle Z, P_{\hat{T}^\perp}(M^*) \right\rangle$$

This is equivalent to Eq. (2), completing the proof. □

## 3 Theoretical Guarantees

Summarizing so far, our estimator is constructed in two steps: solve the convex program in Eq. (1a) to obtain an initial estimate $(\hat{M}, \hat{\tau})$, then de-bias according to Eq. (1b). While we have presented this estimator in a setting that allows for *multiple* treatments (i.e. $k \geq 1$), for the sake simplicity our results here (Theorems 1 and 2) are restricted to the single treatment (i.e. $k = 1$) setting. Recall that the work on synthetic control, which we seek to extend, is for a single treatment and a particular form of $Z_1$ (support on a single row); our results in this section are for a single treatment but *general $Z_1$*. To ease notation, we will from here on suppress treatment-specific subscripts ($Z_1$, $\tau_1$, etc.)

As mentioned earlier, our results require a set of conditions that relate the treatment matrix $Z$ to the tangent space $T^*$ of $M^*$:

---

[5] https://arxiv.org/abs/2106.02780

*Assumption* 3. There exist positive constants $C_{r_1}, C_{r_2}$ such that

(a) $\|ZV^*\|_F^2 + \|Z^\top U^*\|_F^2 \leq (1 - C_{r_1}/\log(n)) \|Z\|_F^2$.

(b) $|\langle Z, U^* V^{*\top}\rangle| \|P_{T^{*\perp}}(Z)\| \leq (1 - C_{r_2}/\log(n)) \|P_{T^{*\perp}}(Z)\|_F^2$.

Assumption 3 is nearly necessary for identifying $\tau^*$ (in a manner made formal by Proposition 2 below). These conditions are mild enough to allow for various treatment patterns that occur in practice, and broadly expand on the set of patterns possible in the synthetic control literature (as we discuss in Section 3.1).

Let $\delta = \mathcal{T} \circ Z - \tau^* Z$ be the matrix of treatment effect 'residuals.' Our first result establishes a bound on the error rate of $\tau^d$. Note that $\delta$ is a zero-mean matrix and is zero outside the support of $Z$. Thus the requirement below that $\|\delta\| \leq C_\delta \sigma \sqrt{n/r}$ is mild. It is trivially met in synthetic control settings. It is also easily seen as met when $\delta$ has independent, sub-Gaussian entries. Finally as it turns out, the condition can also admit random sub-gaussian matrices with complex correlation patterns; see [36].

**Theorem 1** (Optimal Error Rate). *Suppose $\frac{\sigma\sqrt{n}}{\sigma_{\min}} \leq C_1 \frac{1}{\kappa^2 r^2 \log^5(n)}$ and $\|\delta\| \leq C_\delta \sigma \sqrt{n/r}$. Then for any $C_2 > 0$, for sufficiently large $n$, with probability $1 - O(n^{-C_2})$, we have*

$$|\tau^d - \tau^*| \leq C_e \log(n) \max \left( \frac{\sigma}{\|Z\|_F}, \frac{\kappa^3 r n \sigma \log^5(n)}{\sigma_{\min}}, \frac{|\langle P_{T^*}(\delta), P_{T^*}(Z)\rangle|}{\|Z\|_F^2} \right).$$

*Here, $C_e$ is a constant depending (polynomially) on $C_1, C_2, C_{r_1}, C_{r_2}, C_\delta$ (where $C_{r_1}$ and $C_{r_2}$ are the constants appearing in Assumption 3).*

To begin parsing this result, consider a 'typical' scenario in which $\sigma, \kappa, r, \mu = O(1)$, and $\sigma_{\min} = \Omega(n)$. Then Theorem 1 implies that

$$|\tau^d - \tau^*| = \tilde{O}\left(\sigma/\|Z\|_F + |\langle P_{T^*}(\delta), P_{T^*}(Z)\rangle|/\|Z\|_F^2\right). \tag{6}$$

This is minimax optimal (up to $\log n$ factors), as shown below:

**Proposition 1** (Minimax Lower Bound). *For any estimator $\hat{\tau}$, there exists an instance with $\sigma, \kappa, r, \mu = \Theta(1)$, and $\sigma_{\min} = \Theta(n)$, on which, with probability at least $1/3$,*

$$|\hat{\tau} - \tau^*| \geq \max \left(\sigma/\|Z\|_F, |\langle P_{T^*}(\delta), P_{T^*}(Z)\rangle|/\|Z\|_F^2\right).$$

Finally, it is worth considering some special cases under which Eq. (6) reduces further to

$$|\tau^d - \tau^*| = \tilde{O}\left(\sigma/\|Z\|_F\right), \tag{7}$$

which is the optimal rate (up to $\log n$) achievable even when $M^*$ and $\delta$ are known. Any of the following are, alone, sufficient to imply Eq. (7):

*Independent $\delta$:* Independent, sub-gaussian $\delta_{ij}$ with $O(1)$ sub-gaussian norm.
*Synthetic control and block $Z$:* $\|\delta\|_{\max} = O(1)$, $Z$ consists of an $\ell \times c$ block that is sufficiently sparse: $\sqrt{\ell c}(\ell + c) = O(n)$. For comparison, state-of-the-art synthetic control results [13, 37] require the sparser condition $\ell c(\ell + c) = O(n)$ (though that condition enables asymptotic normality results).
*Panel data regression:* The conditions imposed in [27], the most notably that $Z$ be sufficiently dense: $\|P_{T^{*\perp}}(Z)\|_F^2 = \Theta(n^2)$. This recovers their error guarantee (up to $\log$ factors); see Appendix D.

Our second main result establishes asymptotic normality for our estimator. This naturally requires some additional control over the variability of $\delta$. We consider the setting in which the $\delta_{ij} = \mathcal{T}_{ij} - \tau^*$ are independent variables.

**Theorem 2** (Asymptotic Normality). *Suppose each $\delta_{ij}$ is a mean-zero, independent random variable with sub-Gaussian norm $\|\delta_{ij}\|_{\psi_2} = O(1)$. Assume $\kappa = r = \mu = \sigma = O(1)$ and $\sigma_{\min} = \Omega(n)$. Then with probability $1 - O(1/n^3)$,*

$$\tau^d - \tau^* = \frac{\langle E + (\delta \circ Z), P_{T^{*\perp}}(Z)\rangle}{\|P_{T^{*\perp}}(Z)\|_F^2} + O\left(\frac{\log^8(n)}{n}\right).$$

*Consequently,*

$$(\tau^d - \tau^*)/V_\tau^{1/2} \to \mathcal{N}(0, 1), \quad V_\tau = \sum_{ij} P_{T^{*\perp}}(Z)_{ij}^2 \mathrm{Var}(E_{ij} + \delta_{ij} Z_{ij}) \Big/ \left(\sum_{ij} P_{T^{*\perp}}(Z)_{ij}^2\right)^2,$$

*provided that $V_\tau^{1/2} = \Omega(\log^9(n)/n)$.*

Asymptotic normality is of econometric interest, as it enables *inference*. Specifically, inference can be performed using a 'plug-in' estimator $\hat{V}_\tau$ for $V_\tau$, gotten by substituting $\hat{T}$ for $T^*$ and $O - M^d - \tau^d Z$ for $E + (\delta \circ Z)$, where $M^d$ is a de-biased estimator for $M^*$ (see Appendix L). This is a common procedure in the literature (e.g., see [18]), and it is straightforward to show that $\hat{V}_\tau \sim V_\tau$.

**Proof Technique:** The proofs of Theorems 1 and 2 are inspired by recent developments on bridging convex and non-convex formulations for matrix completion [4] and Robust-PCA [5]. Whereas that work assume a random, independent missingness pattern, our proof extends that program to deal with deterministic treatment patterns Z. As such, this analysis is likely of interest, in its own right, as a complement to the matrix completion literature [38, 39, 4, 5]. Broadly, we must address the issue that constructing a dual certificate to analyze the quality of our convex estimator directly is hard. Instead [4] show the existence of such a certificate non-constructively by studying a non-convex estimator and showing that a (fictitious) gradient descent algorithm applied to that estimator recovers a suitable dual certificate. We effectively extend that program to deterministic patterns $Z$, and provide entry-wise recovery guarantees on $M^*$ in this setting that are of independent interest.

### 3.1 Applicability of Treatment Conditions

Having stated our results, we return to our assumptions on $Z$ (Assumption 3) and discuss the extent to which they allow for treatment patterns that occur in practice. First however, we establish that Assumption 3 is necessary. Consider the proposition below, which establishes that should either of the condition of Assumption 3 be violated by an amount that grows negligible with $n$, then identification is rendered impossible so that no estimator can recover $\tau^*$.

**Proposition 2.** *Let $n$ be any positive even integer. There exists a matrix $Z \in \{0,1\}^{n \times n}$ and a pair of rank-1 and $\mu = 2$ (incoherent) matrices $M_1, M_2 \in \mathbb{R}^{n \times n}$ with SVDs denoted by $M_i = U_i \Sigma_i V_i^\top$ and $T_i$ being the tangent space of $M_i$, such that all three of the following statements hold:*

1. $\|ZV_i\|_{\mathrm{F}}^2 + \|Z^T U_i\|_{\mathrm{F}}^2 = \|Z\|_{\mathrm{F}}^2, \quad i = 1, 2$
2. $\left| \langle Z, U_i V_i^\top \rangle \right| \|P_{T_i^{*\perp}}(Z)\| = \|P_{T_i^{*\perp}}(Z)\|_{\mathrm{F}}^2, \quad i = 1, 2$
3. $M_1 + Z = M_2$

We next discuss various treatment patterns that are admissible under Assumption 3:[6]

*1. Rank grows faster than $r$:* Assumption 3 holds if $\sum_{i=1}^r \sigma_i(Z)^2 \leq (1 - C/\log n) \|Z\|_{\mathrm{F}}^2/(\sqrt{r}+2)$. Loosely speaking, this requires that the rank of $Z$ be strictly higher than $r$, and that less than $1/\sqrt{r}$ of its 'mass' lie in its first $r$ components. Put another way, $Z$ must be sufficiently different from any rank-$r$ approximation. One common setting where this occurs is when there is sufficient randomness in generating $Z$, such as the case where the entries of $Z$ are drawn independently, which is the canonical scenario in the matrix completion literature (e.g., [35, 38, 39]).

*2. Maximal number of ones in a row and column:* Let $\ell$ and $c$ denote the maximum number of ones in a row and column, respectively, of $Z$. Assumption 3 holds if $\ell + c \leq (1 - C/\log n) n/(r^2 \mu)$. When $r, \mu = O(1)$, this allows $\ell, c = O(n)$ and generalizes the sparse block patterns studied in the literature (e.g. [40, 13]), where $Z$ is a two-by-two block matrix with exactly one block equal to one.

*3. Single row or column (Synthetic Control):* Consider the case when $Z$ is supported on a single row (or column, equivalently), as in synthetic control. Assumption 3 holds if $\|z^\top V^*\|_{\mathrm{F}}^2 \leq (1 - C/\log n - \mu r/n) \|z\|^2$, where $z^\top$ is the non-zero row of $Z$. This will easily hold, if allowing a negligible perturbation to either $z_1$ or the row space of $M^*$. It is also interesting to note that the identification assumption made in the canonical paper [2] (i.e., $T_0^{-1} \sigma_{\min}(\sum_{i=1}^{T_0} V_i^* V_i^{*\top}) > 0$ is bounded away from zero, where $T_0 = \max_{z_i=0} i$), together with $T_0 = \Omega(n/\log(n))$ and $\mu r = O(n/\log(n))$ (also implicitly assumed in [2] for an optimal gaurantee), together imply Assumption 3.

We end this section by drawing a connection to the literature on panel data regression with interactive fixed effects [25, 36, 27]. That literature studies estimators similar in spirits to ours (e.g., [27] analyzed the performance of the convex estimator and a heuristic de-biasing approach). However,

---

[6]Also see a further discussion for mathematical equivalent forms of Assumption 3 based on basis transformations and the generalization to $k > 1$ in Appendix C.

those approaches are only known to work if $\|P_{T^{*\perp}}(Z)\|_{\mathrm{F}} = \Theta(n^2)$[7]. This is of course a substantially stronger assumption than Assumption 3 and rules out sparse treatment patterns (as in synthetic control). In summary, our approach also has the potential to broaden the scope of problems addressed via panel data regression.

## 4 Experiments

We conducted a set of experiments on semi-synthetic datasets (the treatment is introduced artificially and thus ground-truth treatment-effect values are known) and real datasets (the treatment is real and ground-truth treatment-effect values are unknown). The results show that our estimator $\tau^d$ is more accurate than existing algorithms and its performance is robust to various treatment patterns.

The following four benchmarks were implemented: (i) Synthetic Difference-in-Difference (SDID) [13]; (ii) Matrix-Completion with Nuclear Norm Minimization (MC-NNM) [3] (iii) Robust Synthetic Control (RSC) [10] (iv) Ordinary Least Square (OLS): Selects $a, b \in \mathbb{R}^n, \tau \in \mathbb{R}$ to minimize $\|O - a1^T - 1b^T - \tau Z\|_{\mathrm{F}}^2$, where $1 \in \mathbb{R}^n$ is the vector of ones. It is worth noting that SDID and RSC one apply to traditional synthetic control patterns (*block* and *stagger* below).

**Semi-Synthetic Data (Tobacco).** The first dataset consists of the annual tobacco consumption per capita for 38 states during 1970-2001, collected from the prominent synthetic control study [2] (the treated unit California is removed). Similar to [3], we view the collected data as $M^*$ and introduce artificial treatments. We considered two families of patterns that are common in the economics literature: *block* and *stagger* [3]. Block patterns model simultaneous adoption of the treatment, while stagger patterns model adoption at different times. In both cases, treatment continues forever once adopted. Specifically, given the parameters $(m_1, m_2)$, a set of $m_1$ rows of $Z$ are selected uniformly at random. On these rows, $Z_{ij} = 1$ if and only if $j \geq t_i$, where for block patterns, $t_i = m_2$, and for stagger patterns, $t_i$ is selected uniformly from values greater than $m_2$.

To model heterogenous treatment effects, let $\mathcal{T}_{ij} = \tau^* + \delta_i$ where $\delta_i$ is i.i.d and $\delta_i \sim \mathcal{N}(0, \sigma_\delta)$ characterizes the unit-specific effect. Then the observation is $O = M^* + \mathcal{T} \circ Z$. We fix $\tau^* = \sigma_\delta = \bar{M}^*/5$ through all experiments, where $\bar{M}^*$ is the mean value of $M^*$ [8]. The hyperparameters for all algorithms were tuned using rank $r \sim 5$ (estimated via the spectrum of $M^*$).

Next, we compare the performances of the various algorithms on an ensemble of 1,000 instances with $m_1 \sim \mathrm{Uni}[1, n_1], m_2 = \mathrm{Uni}[1, n_2)$ for stagger patterns and $m_1 \sim \mathrm{Uni}[1, 5], m_2 = 18$ for block patterns (matching the year 1988, where California passed its law for tobacco control). The results are reported in the first two rows of Table 1 in terms of the average normalized error $|\tau - \tau^*|/\tau^*$.

Note that the treatment patterns here are 'home court' for the SDID and RSC synthetic control methods but our approach nonetheless outperforms these benchmarks. One potential reason is that these methods do not leverage all of the available data for learning counterfactuals: MC-NNM and SDID ignore treated observations. RSC ignores even more: it in addition does not leverage some of the *untreated* observations in $M^*$ on treated units (i.e. observations $O_{ij}$ for $j < t_i$ on treated units).

Table 1: Comparison of our algorithm (De-biased Convex) to benchmarks on semi-synthetic datasets (Block and Stagger correspond to Tobacco dataset; Adaptive pattern corresponds to Sales dataset). Average normalized error $|\tau - \tau^*|/\tau^*$ is reported.

| Pattern | De-biased Convex | SDID | MC-NNM | RSC | OLS |
|---|---|---|---|---|---|
| Block | 0.15 (±0.13) | 0.23 (±0.19) | 0.27 (±0.24) | 0.30 (±0.26) | 0.38 (±0.36) |
| Stagger | 0.10 (±0.20) | 0.16 (±0.18) | 0.15 (±0.16) | 0.20 (± 0.27) | 0.18 (± 0.19) |
| Adaptive | 0.02 (±0.02) | - | 0.13 (±0.10) | - | 0.20 (±0.18) |

**Semi-Synthetic Data (Sales).** The second dataset consists of weekly sales of 167 products over 147 weeks, collected from a Kaggle competition [41]. In this application, treatment corresponds to

---

[7]Parenthetically, the assumption $\|P_{T^{*\perp}}(Z)\|_{\mathrm{F}} = \Theta(n^2)$ greatly simplifies analysis in [27] since a global bound on $\tau - \tau^*$ can be easily obtained. One of the main technical innovations in our paper, building on recent advances in the matrix completion literature, is to conduct a refined 'local' analysis without the assumption on the density of $Z$.

[8]See Appendix E for estimating row-specific treatment effects

various 'promotions' of a product (e.g. price reductions, advertisements, etc.). We introduced an artificial promotion $Z$, used the collected data as $M^*$ ($\bar{M}^* \approx 12170$), and the goal was to estimate the average treatment effect given $O = M^* + \mathcal{T} \circ Z$ and $Z$ ($\mathcal{T}$ follows the same generation process as above with $\tau^* = \sigma = \bar{M}^*/5$).

Now the challenge in these settings is that these promotions are often decided based on previous sales. Put another way, the treatment matrix $Z$ is constructed *adaptively*. We considered a simple model for generating adaptive patterns for $Z$: Fix parameters $(a, b)$. If the sale of a product reaches its lowest point among the past $a$ weeks, then we added promotions for the following $b$ weeks (this models a common preference for promoting low-sale products). Across our instances, $(a, b)$ was generated according to $a \in \mathrm{Uni}[5, 25], b \in \mathrm{Uni}[5, 25]$. This represents a treatment pattern where it is unclear how typical synthetic control approaches (SDID, RSC) might even be applied.

The rank of $M^*$ is estimated via the spectrum with $r \sim 35$. See Table 1 for the results averaged over 1,000 instances. The average of $|\tau - \tau^*|/\tau^*$ is $\sim 5.3\%$ for our algorithm, versus $27.6\%$ for MC-NNM. We conjecture that the reason for this is that highly structured missing-ness patterns are challenging for matrix-completion algorithms; we overcome this limitation by leveraging the treated data as well. Of course, there is a natural trade-off here: if the heterogeneity in $\delta$ were on the order of the variation in $M^*$ (so that $\|\delta \circ Z\| \gg \sigma_r(M^*)$) then it is unclear that the treated data would help (and it might, in fact, hurt). But for most practical applications, the treatment effects we seek to estimate are typically small relative to the nominal observed values.

**Real Data.** This dataset consists of daily sales and promotion information of 571 drug stores over 942 days, collected from Rossmann Store Sales dataset [42]. The promotion dataset $Z$ is binary (1 indicates a promo is running on that specific day and store). The *real* pattern is highly complex (see Fig. 1) and hence synthetic-control type methods (SDID, RSC) again do not apply. Our goal here is to estimate the average increase of sales $\tau^*$ brought by the promotion.



| | $\tau$ | Test Error |
|---|---|---|
| De-biased Convex | 118.2 ($\pm$2.4) | 0.04 ($\pm$0.002) |
| MC-NNM | -49.4 ($\pm$0.98) | 0.07 ($\pm$0.002) |
| OLS | -45.8 ($\pm$1.24) | 0.18 ($\pm$0.003) |

Figure 1: *Left:* The promotion pattern of the real data. *Right:* Estimation of $\tau$ and test errors.

The hyperparameters for all algorithms were tuned using rank $r \sim 70$ (estimated via cross validation). A test set $\Omega$ consisting of 20% of the treated entires is randomly sampled and hidden. The test error is then calculated by $\|P_\Omega(M + \tau Z - O)\|_{\mathrm{F}}^2 / \|P_\Omega(O - \bar{O})\|_{\mathrm{F}}^2$ where $\bar{O}$ is the mean-value of $O$. Fig. 1 shows the results averaged over 100 instances. Our algorithm provides superior test error. This is potentially a conservative measure since it captures error in approximating both $M^*$ and $\tau^*$; the variation contributed by $M^*$ to observations is substantially larger that that contributed by $\tau^*$. Now whereas the ground-truth for $\tau^*$ is not known here, the negative treatment effects estimated by MC-NNM and OLS seem less likely since store-wise promotions are typically associated with positive effects on sales.

**Asymptotic Normality.** The normality of our estimator is also verified, where the prediction from Theorem 2 is precise enough to enable inferential tasks such as constructing confidence intervals (CIs) for $\tau^*$: our 95% CIs typically had "true" coverage rates in the range of 93-96% for a synthetic set of instances described in Appendix E. See Fig. 2.

## 5   Conclusion

Motivated by the extremely important econometric problem of estimating treatment effects from panel data, we studied a natural formulation of this problem as one of recovering an unknown quantity

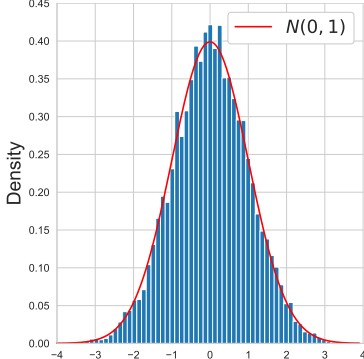

| | $n_1 = 50$ | 100 | 150 | 200 |
|---|---|---|---|---|
| $n_2/n_1 = 0.5$ | 0.916 | 0.957 | 0.956 | 0.942 |
| 1 | 0.953 | 0.946 | 0.954 | 0.939 |
| 2 | 0.946 | 0.947 | 0.945 | 0.957 |
| 4 | 0.94 | 0.934 | 0.949 | 0.944 |

Figure 2: Evaluation of our distributional characterization of $\tau^d$ on a synthetic ensemble where $\delta$ and $E$ follow i.i.d Gaussian distribution. *Left:* Empirical Distribution of $(\tau^d - \tau^*)/V_\tau$ with $n = 100$, overlaid with the $\mathcal{N}(0,1)$ density function as predicted by Theorem 2. *Right:* Coverage rates of $95\%$ confidence intervals (the 'correct' coverage rate is 0.95) for different sizes $(n_1, n_2)$ with $r = 10$ . See Appendix E for the data generation processes in details.

that has been added to an unknown low-rank matrix at a subset of its entries. We proposed an estimator based on solving, and de-biasing, a natural regularized least-squares objective. We built on recent techniques for establishing entry-wise guarantees that leverage a connection between convex and non-convex formulations, and proved that our estimator is order-optimal with nearly minimal conditions on the underlying low-rank matrix and the pattern of 'treated' entries.

One exciting direction for further work is to consider the correlation between noises (as opposed to the independent noise assumption). In fact, our main guarantee for the error rate (Theorem 1) is not restricted to independent noise (the results incorporate a deterministic $E$ naturally, see Appendix B and Appendix J). On the other hand, our asymptotic normality result (Theorem 2) indeed requires independent noise. Generalizing this result to correlated noise appears to be more involved and may require further technical innovations.

## 6   Ackowledgement

We thank all anonymous reviewers for their constructive comments. Farias and Peng were partially supported by NSF Grant CMMI 1727239.

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
