# OpenReview forum: "Learning Treatment Effects in Panels with General Intervention Patterns"
_NeurIPS.cc/2021/Conference — NeurIPS 2021 Oral_

### Official Review · Reviewer_gN9T · 2021-07-16

**Rating:** 9
**Confidence:** 3

**Summary:**

The paper studies learning treatment effects in panel data with heterogenous treatment effects, with a de-biased estimator that achieves rate-optimal recovery.

**Limitations And Societal Impact:**

The theoretical results are only for $k=1$.

**Main Review:**

The paper is well-written and clearly organized. This paper not only extends the synthetic control paradigm to the much broader class, in the sense of robust matrix completion, the paper also delivers novel theoretical results with unknown and deterministic missing patterns. Thus, the reviewer believes that this would be a nice contribution to the Neurips community and also provides new technical tools that could be helpful to other problems. Some minor points:

1. The paper presents the problem under the setting of multiple treatment effects that allows for $k \geq 2$. However, the main theorems are delivered assuming $k = 1$. Although I believe this is definitely reasonable to be left for future work, I wonder whether there are technical barriers or it could be generalized without much work?
2. I wonder whether assumption 3 itself is enough for showing the multiple treatment effect case. It is possible that different treatment effects (say $Z_1, Z_2$) cancels in the tangent space of $M^\star$. Would that be a problem?

**Time Spent Reviewing:**

6

---

> ### Author Response · Authors · 2021-08-10
> **Response to minor points**
>
> Thank you for the positive review. In response to your minor points:
>
> 1. The generalization to $k>1$: There does not appear to be any technical barrier to generalizing the proof framework to $k>1$, although the proof will be more mechanically involved, particularly when performing the inductive analysis under multiple treatment effects.
>
> 2. Assumption 3 when $k>1$: Excellent question! Assumption 3 itself, for each single $Z_i$, would not be sufficient for the $k>1$ case. The reason is exactly as you point out: $Z_1 - Z_2$ may be in the tangent space of $M^{*}$, which would render identification impossible. Roughly speaking, the identification condition we would require is that every linear combination of $Z_i, i=1,2,\dotsc,k$ satisfies Assumption 3 (this could be restrictive in practice when $k$ is large, but one could for example mitigate this by regularization or adding explicit constraints on the $\tau_i$). We will discuss this in the final version.

---

### Official Review · Reviewer_BWiu · 2021-07-16

**Rating:** 10
**Confidence:** 4

**Summary:**

The paper considers the problem of average treatment effect estimation in panel data. The authors consider a setup where the observed outcomes form a matrix with a special structure: it is the sum of a counterfactual matrix and a treatment matrix, where the counterfactual matrix is assumed to be a “noisy observation” of a low rank matrix. The paper proposes a debiased estimator, shows that it is rate optimal, derives its asymptotic normality and demonstrate its performance through numerical experiments.

**Limitations And Societal Impact:**

The authors have adequately addressed the limitations and potential negative societal impact of their work.

**Main Review:**

Originality: The proposed estimator is new. The debiasing technique is very interesting and inspiring. In terms of theory, as far as I know, this is the first rate optimality result in this set up.

Quality: As far as I checked, all claims are well supported.

Clarity: Overall, the paper is well written and I enjoyed reading it. The messages are delivered in a very clear way. I enjoyed a lot reading the proof of Lemma 1. It is very helpful for readers to understand the key idea behind the debiasing technique.

Significance: Overall, I think the paper makes a strong and important contribution. The problem this paper considers is very important. I believe rate optimality is also a very difficulty goal especially under very general assumptions. The paper provides a strong solution to the problem. Both the upper bound and the lower bound are very nice results. I believe the results and proof techniques are not only important in the field of panel data, but also for other matrix estimation/completion problems in general. The proposed estimator appears to perform very good empirically as well.

**Time Spent Reviewing:**

3

---

> ### Author Response · Authors · 2021-08-10
> **Thanks!**
>
> We appreciate your support.

---

### Official Review · Reviewer_55D9 · 2021-07-16

**Rating:** 7
**Confidence:** 3

**Summary:**

This paper considers a general problem of applying multiple treatments (heterogenous) to multiple units across multiple time-steps, where the outcome is of this treatment has a linear relationship. Given the treatment decision matrix and the outcome matrix, the goal of the algorithm is to obtain the treatment effect (or more specifically a sufficient statistic that can be used to derive the treatment effect given the model). In particular, the outcome matrix O for n units across T steps is determined by a linear system M* + E + sum_{k} treatment_effect_matrix_k * decision_matrix_k. The algorithm sees the outcome matrix O and knows the decision_matrix_k for each of the k treatments and it needs to recover the treatment_effect_matrix_k for each of the k treatments. The paper identifies assumptions under which such an estimator is possible, shows that when violating one of the assumptions no algorithm can obtain such an estimator and that they have good empirical performance on semi-synthetic datasets.

**Limitations And Societal Impact:**

One of the limitations of this paper is that of independent noise E. As stated above its not an outrageous assumption, but at the same time by far its the biggest limitation of this model. If we treat the outcome matrix as n*T where n are the units in the system and T is the time-steps, then the noise across the time-steps is rarely independent. In fact, they would be systematic and positively correlated. A lesser issue, but still similar is that across experimental units. The noise across different customers/experimental units will also be correlated. For instance, across two "similar" users, it is likely that the noise (typically used to include all the unmeansurable parts of the system) would also be correlated in a systematic way. Acknowledging this limitation in the paper is useful to the reader/a practioner who may potentially try to use this. Of course, making a reasonable non-independence assumption to make the problem tractable is challenging and is reasonable to assume that is beyond the scope of the current paper.

Regarding societal impact, I think the paper is primarily foundational. So as a stand alone paper does not have negative societal impact. But based on application it could have impact. For instance, if the "treatment" in question is on human subjects, it may require ethics considerations. But that is largely beyond the scope of the current paper.

**Main Review:**

Originality: The problem/model considered in this paper is very general and captures many important problems as special cases: single treatment effect estimation, synthetic control estimator, etc. Thus, as far as I can tell, the problem setting is original. The paper identifies assumptions under which an unbiased estimator is possible. In particular, the important assumption is Assumption 3 (ASsumption 1 is independence of entries of the noise and ASsumption 2 is that of incoherence both are not outrageous assumptions). They show that when this assumption is made an algorithm based on solving convex program can be used to obtain an unbiased estimator for the problem. Moreover, they also show that if this is violated then no algorithm can recover the required statistic. This constitutes a reasonably complete characterization of the problem in my opinion. Although once the assumptions are made, the recovery/proof is pretty straightforward, in my opinion the original contribution of the paper is to identify these set of tracatable assumptions.

Quality/Clarity: The paper is well-written and easy to read. I was able to understand the main ideas of the paper, the proof very easily. I also think the paper does a detailed job (as allowed by the page limit) on experimenting this with semi-synthetic datasets. Overall the results in this paper look complete to me; identify assumptions the estimator, show that its necessary in theory (with some cavets) and experiment the usefulness in some reasonable empiricial setting.

Significance: The paper adds to the literature of estimating treatment effects of which there are many different lines of work. Synthetic control (a special case of their model) has found usefulness in doing counter factual estimation. Their work also can be conctrued to fall in the line of work on linear casual models (although a simpler version since the main challenges in causal inference is to deal with confounders or correlated noise). The particular setting itself seems reasonably practical in large scale settings like recommender systems where one tends to apply multiple treatments to multiple experimental units simultaneously. One could argue if the linearity assumption between outcome and treatment holds but upto that caveat, the setting and the method is significant/practical.

**Time Spent Reviewing:**

7

---

> ### Author Response · Authors · 2021-08-10
> **The independent noise assumption**
>
> Thank you for highlighting the independent noise assumption – we agree that complete independence, particularly across time steps and units, would be limiting. We should clarify that our main guarantee for the error rate (Theorem 1) is not restricted to independent noise. In particular, the only conditions on E required for our proof are that $||E|| \lesssim \sqrt n$, $|<Z,E>| \lesssim \sqrt n||Z||\_{F}$, $|<P_{T^{*\perp}}(Z), E>| \lesssim \sqrt{n/\log n}||Z||\_{F}$ (see Lemmas 3, 4, 5 in Appendix B). These conditions do allow for non-trivial extensions beyond independent noise, e.g., see [26] for a large class of correlated noise models with $||E|| \lesssim \sqrt n$.
>
> On the other hand, our asymptotic normality result (Theorem 2), is indeed for independent noise. Generalizing this result to correlated noise appears to be more involved (a careful examination of the proof suggests that some concentration properties of the rows and columns of E are required), and we leave this for future work.
>
> We will highlight all of this in the final version to clarify the results and limitations (thank you again for the suggestion).

---

### Official Review · Reviewer_KTHy · 2021-07-17

**Rating:** 7
**Confidence:** 3

**Summary:**

This paper looks at the problem of learning treatment effects from observational data. Recently, there has been a flurry of work in establishing theoretical guarantees for low-rank matrix recovery in presence of a Gaussian noise, where de-biasing-based estimators have been analyzed. This paper takes that problem to a next level, in the context of learning treatment effects. Here, the observed matrix is a low-rank matrix added with noise and also treatment matrices. The goal is to recover the effect of average treatment. The authors propose a two-step estimator. In the first step, they try to learn a low-rank matrix and the treatments using a convex optimization program, which is standard in the matrix recovery literature. Then they also use a de-biasing step to improve the performance of their estimation of the treatment matrix.

**Limitations And Societal Impact:**

Yes

**Main Review:**

Finally, they show that their estimator works well in practice for both real and synthetic datasets. They heavily built upon the prior work on low-rank matrix recovery and the new contributions are theoretically and practically justified. I think this is a good paper and vote for acceptance.

**Time Spent Reviewing:**

10

---

> ### Author Response · Authors · 2021-08-10
> **Thanks!**
>
> We appreciate your kind words.

---

### Decision · Program_Chairs · 2021-09-27

**Decision:**

Accept (Oral)

**Comment:**

This paper considers the problem of learning treatment effects with panel data and provides an algorithm with tight theoretical guarantees as well as good empirical performance. All the reviewers agree that the results in this paper are a substantial improvement over existing results for this problem. On the technical side, this problem is closely related to matrix completion but at the same time requires new technical results, which the authors do, building upon recent advances in understanding the convex relaxation formulation of matrix completion through a nonconvex viewpoint. Overall, I feel that this paper has both interesting results as well as techniques that are more broadly interesting.